# Physical Load While Using a Tablet at Different Tilt Angles during Sitting and Standing

**DOI:** 10.3390/s22218237

**Published:** 2022-10-27

**Authors:** Yosuke Tomita, Yoshitaka Suzuki, Akari Shibagaki, Shingo Takahashi, Yoshizo Matsuka

**Affiliations:** 1Department of Physical Therapy, Faculty of Health Care, Takasaki University of Health and Welfare, Takasaki 370-0033, Japan; 2Department of Stomatognathic Function and Occlusal Reconstruction, Graduate School of Biomedical Sciences, Tokushima University, Tokushima 770-8503, Japan; 3Department of Healthcare Informatics, Faculty of Health and Welfare, Takasaki University of Health and Welfare, Takasaki 370-0033, Japan

**Keywords:** ergonomics, kinematics, tablet, electromyography, inertial measurement unit

## Abstract

Few standards and guidelines to prevent health problems have been associated with tablet use. We estimated the effects of posture and tablet tilt angle on muscle activity and posture in healthy young adults. Seventeen healthy young adults (age: 20.5 ± 3 years) performed a cognitive task using a tablet in two posture (sitting and standing) and tablet tilt angle (0 degrees and 45 deg) conditions. Segment and joint kinematics were evaluated using 16 inertial measurement unit sensors. Neck, trunk, and upper limb electromyography (EMG) activities were monitored using 12 EMG sensors. Perceived discomfort, kinematics, and EMG activities were compared between conditions using the Friedman test. The perceived discomfort in the standing-0 deg condition was significantly higher than in the remaining three conditions. Standing posture and tablet inclination significantly reduced the sagittal segment and joint angles of the spine, compared with sitting and flat tablet conditions. Similarly, standing posture and tablet inclination significantly reduced EMG activities of the dorsal neck, upper, and lower trunk muscles, while increasing EMG activity of shoulder flexors. Standing posture and tablet inclination reduced the sagittal flexion angle, and dorsal neck, upper, and lower trunk muscle activities, while potentially increasing the muscle activity of arm flexors.

## 1. Introduction

Information and communication technology (ICT) equipment has been employed in different sectors, such as industry and education. Workers and students use different ICT devices, including desktop computers, laptop computers, tablet devices, and smartphones. Among ICT equipment, mobile electronic devices with touch-screen features—such as tablet computers (tablets) and smart phones—have become increasingly popular worldwide in recent years. According to the annual nationwide survey of the Japanese Ministry of Internal Affairs and Communications in 2020, 95.8% of the young Japanese population aged between 10 and 17 years were using the internet; among them, 70.1% and 37.9% accessed the internet using smart phones and tablets, respectively [1]. The same survey showed that the average time spent browsing the internet was 205.4 min and 73.2 min per day for smartphones and tablets, respectively, highlighting that these users spend a long time using these devices daily. Similar international survey results demonstrated that 63% of the entire population and 71% of the youth aged between 15 and 24 years were using the internet in 2021, with mobile electronic devices being the most common device used [2]. Furthermore, mobile electronic devices are gaining popularity, particularly in education. The Japanese Ministry of Education, Culture, Sports, Science and Technology started a digitalized education program in 2021 (the GIGA school program), where one mobile ICT device—such as a tablet—is allocated to each student [3]. However, few standards and guidelines to prevent health problems associated with habitual tablet use exist; thus, there is a risk of developing neck and shoulder musculoskeletal problems, as discomfort associated with using tablet device is apparent in the neck and shoulders [4]. 

Ergonomic standards and guidelines to prevent health problems—such as the International Organization for Standardization ISO-9241, National standards: Australian Standard (AS-3590.2), Canadian Standard (Can/CSA-Z412-M89), and American Standard (ANSI/HFES-100)—are well established for computer workstation designs, and studies have shown that the height and orientation of the display strongly correlate with muscle strain and discomfort during desktop and laptop computer tasks [5,6,7,8,9]. The sit-stand workstation has been proposed as one of the solutions to address the musculoskeletal problems of the upper limb and neck associated with sedentary work [10]. Lower back muscle activity and lumbar compressive loads were reduced in standing, compared with sitting position [11]. However, to what extent the standards and guidelines for computer workstation designs can be applied to tablet use remains unclear. 

One of the most distinct features of the tablet device from traditional computer devices (e.g., desktop and laptop computers) is that it provides an integrated screen interface, combining both touch panel and display in one central location. This integrated interface makes it difficult for users to adopt the workstation design recommended for computers, as the elevated display forces the users to elevate their arms. One previous study about the effect of tablet tilt angle on physical load reported that the neck flexion angle was reduced when the tablet was tilted for 45 deg, compared to the flat position (0 deg) in sitting, suggesting that physical load on the neck was reduced by steeper tablet tilt angle [4]. However, users preferred to have the tablet flat [12] or with a small tilt angle [4] on the desk when gaming, implying that these preferred tablet positions may lead to increased physical load such as spinal flexion and neck and upper trunk muscle strain. It is therefore important to examine to what extent different postures and screen tilt angles recommended for computer workstations can be applied to tablet use. This study aimed to estimate the effects of posture (sitting vs. standing) and tablet tilt angle (0 deg vs. 45 deg) on physical load (perceived discomfort, joint angle, and muscle activity) in healthy young adults.

## 2. Materials and Methods

### 2.1. Participants

Seventeen healthy young adults (7 females and 10 males) were included (median ± interquartile range, age: 20.5 ± 3 years; height: 172.3 ± 4.7 cm; body weight: 65.7 ± 6.0 kg; body mass index [BMI]: 22.5 ± 2.4 kg/m^2^). Participants were excluded from the study if they reported musculoskeletal injuries and diseases that would limit their ability to perform the task. Participants were instructed to refrain from caffeinated food and drink, drugs, alcohol, tobacco, and any form of nicotine use within 24 h prior to the experiment. After explaining the procedures and risks of the study, written consent was obtained from all participants. The study was approved by the institutional review board (approval number: 3027-2).

### 2.2. Experimental Procedures

The experimental procedures are summarized in Figure 1. Four experimental conditions were implemented, including two postures (sitting and standing) and two tablet tilt angles (0 deg and 45 deg). The order of experimental conditions was randomly allocated for each participant, using a randomization table generated by the Microsoft Excel (Microsoft Corporation, Redmond, WA, USA). An interval of 3 min was set between conditions to avoid fatigue. For the sitting condition, the subject sat on a height-adjustable stool with their hip, knee, and ankle joints flexed at 90 deg (Figure 1A). For the standing condition, the subject stood with their feet shoulder-width apart (Figure 1B).

We used a tablet 247.6 mm × 178.5 mm in size, and 466 g in weight (iPad Pro 11-inch; Apple Inc., Cupertino, CA, USA). The tablet was firmly stabilized in the portrait orientation using a metal tablet stand. In the 0 deg condition, the tablet was fixed parallel to the floor (Figure 1C,D); in the 45 deg condition, the tablet was inclined to 45 deg (Figure 1E,F). In both the 0 deg and 45 deg conditions, the lower edge of the tablet was matched with the position of the middle finger when the participant flexed the elbow 90 deg without flexing the shoulder (a and b in Figure 1A,B). Participants performed the Symbol Digit Modalities Test (SDMT), where they substituted a number using a tablet stylus pen (Apple pencil; Apple Inc., Cupertino, CA, USA) for randomized presentations of geometric figures. The SDMT was performed for 90 s in each experimental condition, and the total number of correct answers was used as the performance outcome of the test.

Perceived discomfort was assessed using the Numerical Rating Scale (NRS); a perceived discomfort of zero was defined as no physical burden, while 10 indicated the greatest physical burden. Participants were asked to verbally report their perceived discomfort level immediately after measurement for each condition.

Kinematic data were acquired using 16 inertial measurement unit (IMU; myoMOTION; Noraxon, Scottsdale, AZ, USA) sensors at a sampling rate of 100 Hz. The IMU sensors were attached to standardized locations on the head, upper trunk, lower trunk, pelvis, both upper arms, forearms, thighs, shanks, and feet (Figure 2A). We used the segment pitch angles (head and upper trunk) and joint angles (neck flexion, thoracic flexion, lumbar flexion, and shoulder flexion) as the kinematic outcomes (Figure 2B). The EMG and IMU systems were synchronized using an electrical synch signal and integrated to the software (MR3; Noraxon, Scottsdale, AZ, USA).

Skin surface areas of all participants were shaved of hair and cleaned with alcohol swabs, and self-adhesive disposable electrodes (G207; Nihon Kohden, Tokyo, Japan) were attached; the distance between paired electrodes was 20 mm. Twelve electromyography (EMG) sensors (Ultium; Noraxon, Scottsdale, AZ, USA) were placed on the muscle belly of the bilateral anterior deltoid (AD), splenius capitis (SPL), upper trapezius (UT), middle trapezius (MT), erector spinae (ES), and multifidus (MUL), based on the recommendation reported in a previous study [13]. Each EMG sensor had an embedded ground electrode, and the EMG signals were collected at a sampling rate of 2000 Hz.

### 2.3. Data Analysis

Segment angles in the global coordinate (head and upper trunk) and the relative angles of body segments (i.e., joint angle; neck flexion, thoracic flexion, lumbar flexion, and shoulder flexion) were calculated by the data acquisition software (MR3; Noraxon, Scottsdale, AZ, USA). The recorded EMG signals were demeaned, rectified, bandpass filtered between 10 and 500 Hz, and smoothened via the Butterworth low-pass filter (2nd order, cutoff frequency of 6 Hz) to produce the linear envelope of the EMG. EMG activity levels were normalized (nEMG) using the peak EMG values of each recorded muscle across all conditions [14,15]. The mean segment and joint angles and mean nEMG activity levels were computed by calculating the mean value of these signals for 60 s during each task condition; data of the first and last 15 s were eliminated from the analysis (Figure 3).

For statistical analysis, the Shapiro–Wilk test was first performed to verify the normal distribution of variables and they did not follow the normal distribution. Therefore, non-parametric tests were used in the subsequent statistical analyses. Condition-specific differences in perceived discomfort, mean segment and joint angles, and mean nEMG activity levels were first compared using the Friedman test. The magnitude and agreement of the difference between conditions were estimated using Kendall’s W, and the Wilcoxon signed-rank test with Bonferroni’s correction was used as the post hoc test for pairwise comparisons between two postural (i.e., sitting vs. standing) and two tablet tilt angle (45 deg vs. 0 deg) conditions. The 95% confidence intervals (CIs) of the median differences between conditions were estimated using the Hodges-Lehman estimate. The relationship between perceived discomfort and quantitative variables (i.e., nEMG activity levels and angle data) was estimated using the Spearman’s correlation coefficient. SPSS software version 22 (IBM, Armonk, NY, USA) was used for statistical analyses, and *p*-values < 0.05 were considered statistically significant.

## 3. Results

The perceived discomfort is summarized in Table 1. The perceived discomfort in the standing-0 deg condition was significantly higher than in the remaining three conditions (χ^2^ = 13.796, *p* = 0.003). Mean segment and joint angles, and mean nEMG activity levels during the task in each experimental condition are summarized in Table 2 and Table 3, respectively.

The changes of kinematics and nEMG activities over time are shown in Figure 3. Mean angles (deg; Table 2) in the sitting position were significantly greater than in the standing position for trunk inclination (median of the difference [lower and upper limit of the 95% CI], 45 deg: 12.2 [8.9, 15.2]; 0 deg: 12.4 [5.7, 18.9]), neck flexion (45 deg: 11.9 [7.5, 15.7]; 0 deg: 11.2 [5.0, 17.2]), lumbar flexion (45 deg: 19.2 [9.8, 28.5]; 0 deg: 19.0 [10.0, 28.7]), and shoulder flexion (45 deg: 12.4 [7.8, 17.3]; 0 deg: 17.5 [10.2, 22.6]) angles. The comparison between different tablet tilt angles revealed that mean angles were significantly greater in the 0 deg than 45 deg condition for the head inclination (sitting: 25 [20.1, 30.0]; standing: 24.2 [20.5, 28.2]), trunk inclination (sitting: 14.3 [10.7, 18.9]; standing: 15.4 [11.6, 19.0]), neck flexion (sitting: 9.9 [6.8, 12.8]; standing: 8.8 [7.7, 10.4]), thoracic flexion (sitting: 4.6 [2.8, 7.2]; standing: 5.5 [3.8, 7.0]), and lumbar flexion (sitting: 7.6 [1.5, 13.6]; standing: 7.4 [4.8, 10.4]) angles.

The mean nEMG activity levels (% peak EMG activity; Table 3) in the sitting position were greater than in the standing position for the right AD (45 deg: 4.8 [0.4, 9.1]; 0 deg: 6.4 [2.8, 10.5]), right SPL (45 deg: 2.8 [0.5, 5.3]; 0 deg: 5.1 [2.3, 9.0]), left SPL (45 deg: 3.3 [1.6, 5.3]; 0 deg: 3.0 [−0.5, 6.0]), right UT (45 deg: 6.0 [3.1, 9.7]; 0 deg: 11.1 [5.7, 16.6]), left UT (45 deg: 2.1 [0.9, 3.6]; 0 deg: 1.8 [−0.5, 1.9]), right MT (45 deg: 6.2 [0.6, 11.3]; 0 deg: 7.3 [3.4, 11.5]), left MT (45 deg: 3.5 [−1.3, 12.1]; 0 deg: 8.7 [2.4, 15.0]), right ES (45 deg: 9.2 [0.4, 14.9]; 0 deg: 7.4 [−0.6, 12.5]), and left ES (45 deg: 11.3 [4.8, 20.9]; 0 deg: 12.5 [6.5, 20.3]). The comparison between tablet tilt angles showed that mean nEMG levels were greater in the 0 deg than 45 deg condition in the left SPL (sitting: 6.9 [3.8, 11.4]; standing: 8.2 [3.9, 12.0]), left UTP (sitting: 4.3 [1.9, 8.6]; standing: 4.5 [2.3, 8.2]), left MTP (sitting: 8.2 [2.8, 14.8]; standing: 4.3 [0.4, 11.9]), right MUL (sitting: 5.9 [−3.9, 14.2]; standing: 12.4 [5.5, 20.1]), and left MUL (sitting: 6.7 [−1.4, 13.0]; standing: 9.9 [2.9, 18.4]), while the mean nEMG activity levels were significantly greater in the 45 deg than 0 deg condition for the right deltoid (sitting: 15.1 [11.9, 19.2]; standing: 17.6 [14.6, 20.6]).

The Spearman’s correlation coefficient (r_s_) for the relationship between perceived discomfort and quantitative variables ranged from −0.204 to 0.180 for mean nEMG activity (*p* > 0.05), and from −0.233 to 0.183 for the segment and joint angles (*p* > 0.05).

## 4. Discussion

We investigated the effects of posture and tablet tilt angles on physical load in healthy young adults. We estimated the physical load using perceived discomfort, joint angles and EMG activities. Our results demonstrated that sagittal head and spine angles were closed to neutral during the standing and 45 deg tablet tilt conditions than during the sitting and 0 deg tablet tilt conditions. EMG activities of the dorsal neck and upper and lower back muscles were greater during the sitting and 0 deg tablet tilt conditions than during the standing and 0 deg tablet tilt conditions. These results suggest that both posture (sitting vs. standing) and tablet tilt angle (45 deg vs. 0 deg) influenced the physical load in healthy young adults.

Standing posture reduced segment and joint angles, as well as EMG activities in the dorsal neck and upper and lower trunk muscles when compared with sitting. These results correspond with those of previous studies demonstrating that EMG activities of the lower back muscles [11], neck muscles [16], and spinal angles (neck, thoracic flexion, and lumbar flexion [16,17,18]) significantly increased during the sitting condition when compared with standing. Furthermore, the benefit of standing versus sitting may be prominent when the activity involves a tablet device. One study showed that postural configurations did not differ between sitting and standing when the task involved a desktop computer [19]; the same study reported that tablet use resulted in a greater neck flexion angle the in sitting position (16.9 deg; 95% CI: 12.8, 21.0), highlighting that performing tablet tasks while sitting can result in a slouched posture. Conversely, our results suggest that the neck flexion angle reduced by 11.9 deg in the 45 deg condition, and 11.2 deg in the 0 deg condition while standing compared with sitting. Similar reductions were also observed in the thoracic and lumbar flexion angles, which were accompanied by reduced dorsal neck and upper and lower back EMG muscle activities. Therefore, the standing posture may be more beneficial for reducing physical load than the sitting posture.

Tablet tilt angle also influenced the body configuration and muscle activities. Studies have shown that sagittal spinal angles and upper back EMG activities were higher during tablet use than during a desktop computer activity, as neck and trunk sagittal angles increased during the tablet task [8]. Previous studies using desktop computers reported that a low monitor location can lead to excessive head flexion relative to the neck [5,6], increasing dorsal cervical muscle activity [7,20]. A monitor location 15–45 deg below horizontal eye level seems to be the most widely recommended monitor position and orientation. One study using a tablet device also reported that 15 deg inclination of the device reduced the neck flexion angle [21]. 

Our study results revealed that sagittal spinal angles and EMG activities in the dorsal neck, upper and lower trunk muscles increased when the tablet was placed at 0 deg (i.e., parallel with floor), which often occurs when the device is placed flat on a desk. Therefore, our results showed that tablets should be used with some inclination to avoid excessive load on the spine, as well as neck, upper and lower trunk muscles. However, it should be noted that inclination of the tablet increased the shoulder flexion angle and deltoid muscle activity on the dominant side (i.e., involved side). Increased deltoid muscle activity seems to be the only downside of tablet inclination; participants reported no significant discomfort associated with an increased tablet tilt angle, as evidence by observing the greatest discomfort level in the standing-0 deg condition. Although the increased deltoid muscle activity observed with tablet inclination may not affect the subjective physical burden during a short-term tablet activity, additional arm support mechanisms (e.g., placement of the forearm on a desk) may reduce the deltoid muscle activity associated with tablet inclination during a long-term tablet task [22]. The weak relationship between perceived discomfort and quantitative variables in our study may be also explained by the fact that the discomfort during tablet tasks can be influenced not only by the physical load associated with spine posture but also by the physical load associated with lifting the arm. Future study needs to investigate the effects of the work environment on different aspects of physical load and discomfort, using tasks that require longer task duration.

This study had several limitations; first, we did not normalize the EMG signal using the maximum voluntary contraction; we had several participants with missing MVC trial data, as the EMG sensors dropped during the MVC measurement trial. We used peak EMG activity levels across all conditions to normalize the EMG signals in each muscle, which is reported to be a reliable method to normalize the EMG signals when within-subject comparison is the main interest of statistical analysis [15]. Second, our study only revealed the short-term effects of posture and tablet tilt angle on physical load; the long-term effects should be examined in future studies using a tablet task with a long task period. Last, we only investigated two conditions for the tablet tilt angle (i.e., 45 deg and 0 deg) while performing a single task (SDMT). It remains unknown which tablet tilt angle is optimal for different tablet tasks, and further studies with different tablet tasks are necessary to establish evidence for the optimal environment for tablet use.

## 5. Conclusions

We investigated the effects of posture (i.e., sitting vs. standing) and tablet tilt angle (0 deg vs. 45 deg) on muscle activity and posture in healthy young adults. Standing posture and inclination of the tablet substantially reduced the sagittal spine flexion angle, as well as dorsal neck and upper and lower trunk muscle activities. However, muscle activity to elevate upper limbs increased when the tablet angle was inclined. Additional upper limb supporting mechanisms may therefore maximize the benefit of a standing posture and inclined tablet angle.

## Figures and Tables

**Figure 1 sensors-22-08237-f001:**
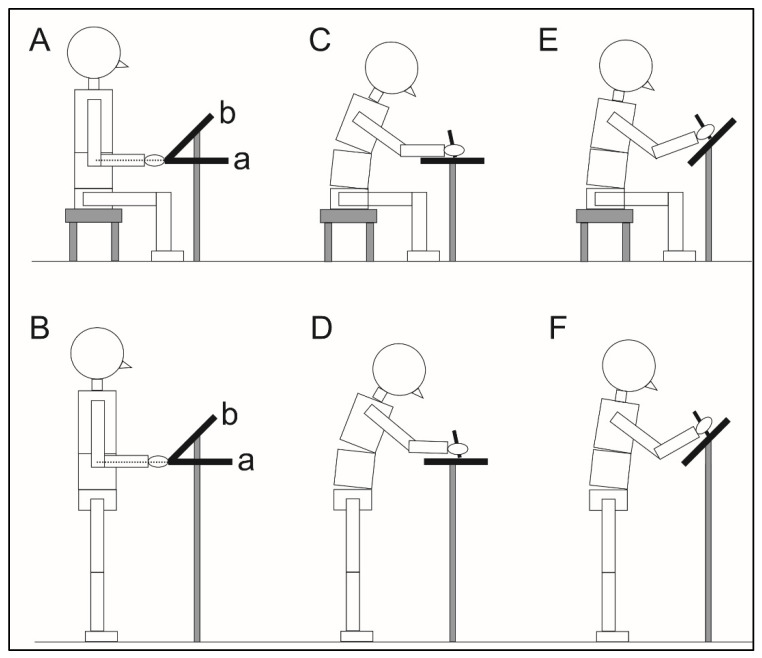
Experimental conditions. In the sitting condition, the subject sat on a height-adjustable stool with their hip, knee, and ankle joint flexed at 90 degrees (deg) (**A**,**C**,**E**). In the standing condition, the subject stood with their feet shoulder-width apart (**B**,**D**,**F**). In both the sitting and standing conditions, the tablet was firmly stabilized in the portrait orientation using a metal tablet stand; in the 0 deg condition, the tablet was fixed parallel to the floor (a in panels (**A**,**B**)), while in the 45 deg condition, the tablet was inclined to 45 deg and the lower edge of the device was matched with the 0 deg condition (b in panels (**A**,**B**)).

**Figure 2 sensors-22-08237-f002:**
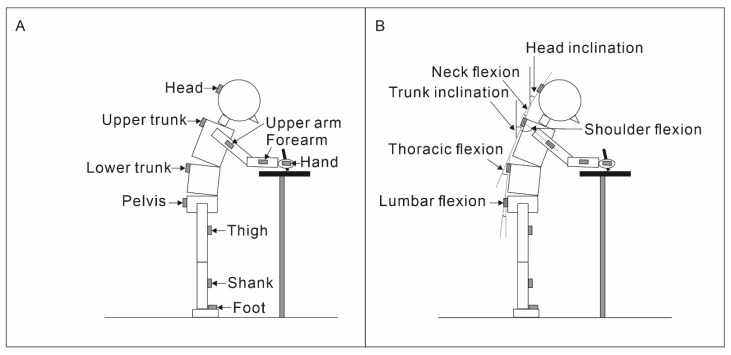
Experimental setup of inertial measurement unit (IMU) data. Sixteen IMU sensors were attached to standardized locations on the head, upper trunk, lower trunk, pelvis, both upper arms, forearms, thighs, shanks, and feet (**A**). Segment angles were measured using the sagittal inclination of each sensor (dashed lines) relative to the global reference (solid lines; **B**), while joint angles were measured using the sagittal inclination of each sensor relative to another sensor (dashed lines; **B**).

**Figure 3 sensors-22-08237-f003:**
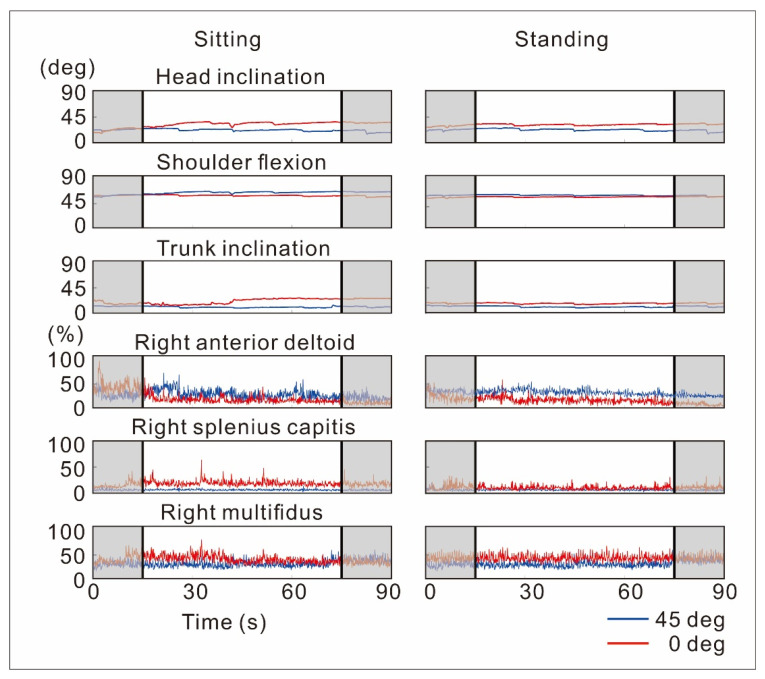
Segment and joint angles, and normalized EMG activities during the task in one participant. Data during 45 deg (blue lines) and 0 deg (red lines) tablet tilt conditions are shown separately for sitting (left column) and standing (right column) conditions. The upper three rows show segment and joint angles (deg). The bottom three rows show normalized EMG activities (%). Data of the first and last 15 s (gray-shaded area separated with black vertical lines) were excluded from the analysis.

**Table 1 sensors-22-08237-t001:** Task performance and subjective physical burden during the task.

	Sitting		Standing		χ^2^ Value (Kendall’s W)	*p*-Value	Post Hoc Test Result
	45 Deg	0 Deg	45 Deg	0 Deg			
Perceived discomfort	3.0 ± 3.0	3.0 ± 3.0	3.0 ± 3.0	4.0 ± 2.0	13.796 (0.271)	0.003	†

† A significant difference was observed between 45 deg and 0 deg conditions.

**Table 2 sensors-22-08237-t002:** Mean segment and joint angles (deg) during the task for each experimental condition.

	Sitting		Standing		χ^2^ Value (Kendall’s W)	*p*-Value	Post Hoc Test Result
	45 Deg	0 Deg	45 Deg	0 Deg			
Head inclination	15.2 ± 6.6	40.3 ± 14.9	15.2 ± 6.4	39.5 ± 13.8	40.835 (0.801)	<0.001	†
Trunk inclination	16.6 ± 10.8	32.8 ± 14.3	4.9 ± 4.2	18.8 ± 12.9	39.988 (0.784)	<0.001	* †
Neck flexion	0.8 ± 10.0	8.8 ± 12.8	10.8 ± 8.1	18.5 ± 10.6	36.035 (0.707)	<0.001	* †
Thoracic flexion	2.3 ± 5.2	5.8 ± 8.7	2.8 ± 4.7	6.7 ± 9.8	28.694 (0.563)	<0.001	†
Lumbar flexion	18.1 ± 24.2	27.3 ± 21.0	3.3 ± 4.8	8.9 ± 9.6	35.612 (0.698)	<0.001	* †
Shoulder flexion	51.9 ± 8.3	47.9 ± 16.1	39.2 ± 13.4	28.8 ± 11.3	31.800 (0.624)	<0.001	*

Values indicate median ± interquartile range. * A significant difference was observed between sitting and standing conditions. † A significant difference was observed between 45 deg and 0 deg conditions.

**Table 3 sensors-22-08237-t003:** Mean normalized muscle activities (% peak EMG activity) during the task in each experimental condition.

	Sitting		Standing		χ^2^ Value (Kendall’s W)	*p*-Value	Post Hoc Test Result
	45 Deg	0 Deg	45 Deg	0 Deg			
**Right**							
Anterior deltoid	37.0 ± 13.4	22.7 ± 16.3	30.1 ± 14.9	13.7 ± 5.1	40.553 (0.795)	<0.001	* †
Splenius capitis	29.1 ± 18.3	31.9 ± 16.1	25.0 ± 13.3	26.1 ± 14.0	19.235 (0.377)	<0.001	*
Upper trapezius	25.5 ± 12.0	25.8 ± 9.8	21.6 ± 13.5	22.3 ± 17.8	13.941 (0.273)	0.003	*
Middle trapezius	31.8 ± 17.6	29.8 ± 7.8	25.3 ± 19.8	23.7 ± 18.5	14.435 (0.283)	0.002	*
Erector spinae	37.7 ± 14.5	36.8 ± 22.1	28.3 ± 17.9	28.2 ± 22.1	11.521 (0.226)	0.009	*
Multifidus	38.5 ± 15.7	52.8 ± 29.6	32.4 ± 13.3	50.1 ± 18.3	15.706 (0.308)	0.001	†
**Left**							
Anterior deltoid	14.7 ± 8.5	17.4 ± 10.1	17.4 ± 6.1	18.6 ± 9.1	5.400 (0.106)	0.145	-
Splenius capitis	34.1 ± 9.4	45.4 ± 12.5	31.3 ± 9.8	37.6 ± 12.5	25.165 (0.493)	<0.001	* †
Upper trapezius	22.7 ± 22.9	27.4 ± 25.5	19.9 ± 22.0	25.6 ± 23.5	25.165 (0.493)	<0.001	* †
Middle trapezius	25.7 ± 21.6	30.0 ± 31.1	20.7 ± 17.2	24.5 ± 18.0	18.459 (0.362)	<0.001	* †
Erector spinae	32.1 ± 19.9	35.9 ± 11.3	19.7 ± 7.5	22.2 ± 10.6	17.500 (0.486)	0.001	*
Multifidus	28.0 ± 15.8	37.6 ± 16.2	16.8 ± 9.6	33.1 ± 23.3	8.976 (0.178)	0.035	†

Values indicate median ± interquartile range. * A significant difference was observed between sitting and standing conditions. † A significant difference was observed between 45 deg and 0 deg conditions.

## Data Availability

Data available on request due to ethical restrictions.

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
