# Peer review of "Physical Load While Using a Tablet at Different Tilt Angles during Sitting and Standing"

_sensors, 2022, doi:10.3390/s22218237_

Round 1

Reviewer 1 Report

Review of the article entitled “Physical load while using a tablet at different tilt angles during sitting and standing”

The authors took up a interesting research topic.

The purpose of this study was to estimate the effects of posture (sitting vs. standing) and tablet tilt angle (0 degrees [deg] vs. 45 deg) on muscle load and posture in healthy young adults.

In this study the results of 17 participants were analyzed.

I have no comments on the Introduction. A short introduction provides a clear background to the work. Properly captured the importance of the problem. Although I have doubts about the purpose of the work. I wonder why the tablet position was selected for the analysis at an angle of 45 degrees.

The methodology has been correctly described in the article, though I have a few insights developed later in the review.

The presentation of the results obtained is clear. The results are presented in the tables.

The discussion was conducted correctly, although in my opinion it could be extended.

The bibliography contains a 19 articles. The bibliography is sufficient, although it could be extended.

My comments:

1)      Please consider the terms "physical loads" and "muscle loads". I think these terms should be defined in the article.

2)      Why is 45 degrees selected? Is this the angle of the most common tablet stands? Please explain?

3)      Please add according to which guidelines the EMG electrodes were placed.

4)      I read at the end of the discussion that the authors did not perform MVC tests for the recorded electromyographic signal for selected muscles.

Has the EMG signal been normalized in some other way?

If no, how was muscle activation compared between persons? After all, muscle activation may vary from person to person, depending on skin preparation, etc.

Comparing the results of muscle EMG activation without any standardization is a big mistake that for this reason the authors obtained very large standard deviations of the results in Table 3. Please reply.

5)      I am asking for a broader explanation of the statement in the discussion:

“However, the impact of the lack of EMG normalization is minimal in our study, as we only included within-subject comparisons using nonparametric tests.”

6)      Are the authors able to estimate this "minimal" impact?

7)      There is no information in the article about checking the normality of the distribution of the analyzed variables - please add.

8)      Line 236-238: „Our results demonstrated that sagittal segment and joint angles were greater during the sitting and 0 deg tablet tilt conditions than during the standing and 45 deg tablet tilt conditions.” This is quite an obvious statement ...

Reviewer 2 Report

The manuscript by Tomita et al. estimated the effects of posture and tablet tilt angle on muscle load and posture in healthy young adults. Segment and joint kinematics were also evaluated using inertial measurement unit sensors, and while neck, trunk, and upper limb electromyography activities were monitored using EMG sensors. Perceived discomfort as well as the quantitative measurements were then compared between conditions. Overall, the manuscript is well written and easy to follow. This reviewer only has a few minor comments.

1. It would be nice to see some of the raw traces with respect to the quantitative measurements.

2a. Are there any correlations between the perceived discomfort and the quantitative measurements?

2b. Would it be possible to compare all the quantitative measures in a single statistical model to see which may be contributing more/most to the perceived discomfort?

Round 2

Reviewer 1 Report

Many thanks to the authors for their detailed answers.

The most important remark was the lack of normalization of the EMG signal. I am glad that the authors have improved the obtained results.

I have no comments on the submitted version of the article. I accept the manuscript for publication in a revised form.